# The Novel Technique of Uniportal Endoscopic Interlaminar Contralateral Approach for Coexisting L5-S1 Lateral Recess, Foraminal, and Extraforaminal Stenosis and Its Clinical Outcomes

**DOI:** 10.3390/jcm10071364

**Published:** 2021-03-26

**Authors:** Ji Yeon Kim, Hyeun Sung Kim, Jun Bok Jeon, Jun Hyung Lee, Jun Hwan Park, Il-Tae Jang

**Affiliations:** 1Department of Neurosurgery, Spine Center, Leon Wiltse Memorial Hospital, Anyang 14112, Korea; endospinekim@gmail.com; 2Department of Neurosurgery, Spine Center, Nanoori Gangnam Hospital, Seoul 06048, Korea; godisma@gmail.com (J.B.J.); nanooriresearch@naver.com (I.-T.J.); 3Department of Internal Medicine, Chosun University School of Medicine, Gwangju 61453, Korea; pp3614@naver.com; 4The Faculty of Medicine of the University of Debrecen, Nagyerdei krt. 94, 4032 Debrecen, Hungary; yyea7133@gmail.com

**Keywords:** endoscopy, lumbar, contralateral, foraminotomy, stenosis, dysesthesia, dorsal root ganglion

## Abstract

Background: Multifocal intra-and-extraspinal lumbar stenotic lesions could be decompressed with one endoscopic surgical approach, which has the advantages of functional structure preservation, technical efficacy, and safety. Methods: A retrospective study was performed on 48 patients who underwent uniportal endoscopic contralateral approach due to coexisting lateral recess, foraminal, and extraforaminal stenosis at the L5-S1 level. Foraminal stenosis grade and postoperative dysesthesia (POD) were analyzed. Visual analog scale (VAS) pain scores, modified Oswestry Disability Index (ODI) scores, and MacNab criteria for evaluating pain disability and response were analyzed. Results: The foraminal stenosis grade of the treated spinal levels was grade 1 (*n* = 16, 33%), grade 2 (*n* = 20, 42%), and grade 3 (*n* = 12, 25%). The rate of occurrence of POD grade 2 and above, which may be related to intraoperative dorsal root ganglion (DRG) retraction injury, was revealed to be 4.2% (two with grade 2, none with grade 3). The patients showed favorable clinical outcomes. Conclusions: Uniportal endoscopic interlaminar contralateral approach is an effective procedure to resolve combined stenosis (lateral recess, foraminal, and extraforaminal region) with one surgical approach at the L5-S1 level. It may be a minimal DRG retracting and facet joint preserving procedure in foraminal and extraforaminal decompression.

## 1. Introduction

Lumbar spinal stenosis is usually due to degenerative changes in older individuals, and it is commonly diagnosed, given better access to advanced imaging [1]. Lumbar spinal stenosis tends to have multiple nerve root compression points [2,3], with paramedian ligamentum flavum hypertrophy and herniation of the nucleus pulposus (HNP) leading to lateral recess stenosis, foraminal exiting nerve root (ENR) compression by foraminal HNP, syndesmophytes, and facet joint hypertrophy, thereby decreasing the anteroposterior diameter and/or overriding the superior articular process (SAP), and thus reducing the foraminal height due to pedicle impingement and extraforaminal ENR compression with HNP. L5 radiculopathy can originate from L4-L5 stenosis and L5-S1 foraminal and extraforaminal L5 nerve root compression pathologies. Currently, microscopic, and biportal endoscopic techniques usually require two separate approaches to resolve lateral recess, foraminal, and extraforaminal compression of the nerve root.

Asymmetrical multifocal lumbar stenosis with correlated clinical and radiological findings is suitable for uniportal full endoscopic contralateral decompression, which has the benefit of being one approach to address multiple points of compression that is less invasive [4,5,6,7,8]. This contralateral approach might have more benefits at the L5-S1 level, because the L5-S1 level has several anatomical features that should be considered during the transforaminal endoscopic approach, the high iliac crest, inclination of disc space, and the wide facet joint overlapping the disc space [9]

In the present study, we performed uniportal endoscopic treatments for contralateral coexisting lateral recess stenosis, foraminal and extraforaminal stenosis via one direction interlaminar contralateral approach to resolve multiple points of compression. We aimed to assess the clinical results, safety, and efficacy of interlaminar contralateral endoscopic lumbar foraminotomy (ICELF) for multifocal compression lesions.

## 2. Materials and Methods

### 2.1. Study Patients

This was a retrospective cohort study of 48 patients who underwent ICELF (Figure 1a,b) due to coexisting lateral recess, foraminal, and extraforaminal stenosis between January 2018 and January 2020 at a single center. A single senior spine surgeon performed all the procedures. All consecutive patients who met the criteria were included in the study within the study period. The patients were followed up for 6 to 24 months, and data were collected during the follow-up period. The patients in the ICELF group met the following inclusion criteria:

Patients with radiating root pain, back pain, and neurogenic claudication and a minimum of 6 weeks of failed conservative treatment.Patients with L5 radiculopathy due to foraminal and extraforaminal stenosis at the L5-S1 level confirmed by MRI (Figure 1c,d).Patients with accompanying lateral recess stenosis at the same side with foraminal pathology, which simultaneously compress the L5 and S1 roots (Figure 1c,d).Patients with stable isthmic or degenerative spondylolisthesis, with less than 10% ventral displacement on dynamic radiography.

We excluded patients due to the following:Lumbosacral transitional vertebra with stenosis at the L5-S1 level.Other lumbar operations (fusion, decompression, discectomy, multi-level ICELF) at different levels performed simultaneously (patients with previous decompression or discectomy at the index level were included.)Accompanying lumbar lateral recess stenosis inducing L5 nerve root compression at the L4-L5 level.

### 2.2. Surgical Procedures

We performed ICELF to treat coexisting lateral recess, foraminal, and extraforaminal stenosis with one direction approach (Figure 1a–d). We provided information to each patient about ICELF and its potential risks and benefits. The information was weighted in the unnecessary decompression and disruption of the anatomy of the ipsilateral unaffected side. Intimate working with the dura can lead to a dural tear. Meanwhile, it is weighted in benefits of minimal invasiveness as multifocal compressions could be resolved with one direction approach. When the patient accepted the operative approach, he or she was included in the present study.

The ICELF approach with channel switching using two different endoscopes was previously described by Kim et al. [4,5,6] as a channel-in-channel technique. This procedure was performed under sedated epidural anesthesia. The patient was placed in a prone position with the spine in flexion. A skin incision was made just lateral to the outer border of the interlaminar space contralateral to the side of the foramen to be decompressed (Figure 1a). A guidewire was seated over the upper part of the lower lamina, and sequential dilatation was achieved up to a diameter of 16 mm. A final working channel of 16 mm was placed and medially angulated for the visualization of the contralateral foramen. We used an endoscopy system with an endoscope with a 15° viewing angle, 10-mm outer diameter, and 6-mm working diameter for central and lateral recess decompression. Bilateral spinolaminar junction was drilled for undercutting the base of spinous process using a 4.5-mm endoscopic drill. After identifying the Y-shaped proximal origin of ligamentum flavum (LF), the sublaminoplasty was performed by drilling out of the inner cortex and partially cancellous bone toward the contralateral side to obtain the working space for the procedure (Figure 2a). Sublaminoplasty was extended craniolaterally until the LF edge is freed to open the foramen and caudally to the superior articular process (SAP) medial border for lateral recess decompression. The LF was initially preserved to protect the dura during the sublaminoplasty, was then removed piecemeal starting from the midline. Additional drilling and punching of the medial part of the inferior articular process (IAP) and SAP were performed to expose the medial foraminal part (Figure 2b).

The endoscopy system was then changed to one with a 251-mm-long endoscope with a 30° viewing angle, 7.3-mm outer diameter, and 4.7-mm working diameter. The smaller diameter and long length endoscope were critical to perform the safe and efficient decompression in the narrowed foramen preventing nerve root retraction. The foramen was entirely exposed after the drilling of the overriding SAP and removal of attached LF (Figure 2c–f); foraminal herniated disc, epidural fat, and compressed ENR were found (Figure 2g). The foramen was explored using a scope with the further drilling of ventral part of the SAP, foraminal discectomy, and syndesmophytes resection (Figure 3a,b). As the foramen enlarged, the free space between the nerve root and extraforaminal herniated disc was opened (Figure 3c). This obtained space was a key route to reach the extraforaminal part and prevented an increase of the intracanal pressure in the narrowed foramen during manipulation of the extraforaminal disc fragments. The endoscope was introduced more in-depth into the extraforaminal space to perform the further extraforaminal decompression (Figure 2d,f). The remaining syndesmophytes compressing the ENR were drilled out before the operation finished (Figure 2e).

All the procedures during foraminal and extraforaminal decompression were performed without ENR retraction. The endpoint was the far lateral area where the exiting root began to curve downward with natural angulation (Figure 2f). Adequate decompression was assessed by confirming the pulsating nerve roots, the anemic nerve root turning pink, and the return of natural angulation of ENR under direct endoscopic visualization (Figure 2g).

### 2.3. Data Collection

Information on patient characteristics, including age, sex, and clinical symptoms including postoperative dysesthesia (POD) and leg weakness, was collected. Furthermore, the nature of the surgery and the operation levels were documented, as were degenerative pathologies and postoperative complications. Ward physicians collected the following information for each patient preoperatively, 1 and 3 months postoperatively, and at the final follow-up. Almost all patients were discharged within 7 days after the operation, but the patients with dural tears were discharged within 14 days after confirming no delayed cerebrospinal fluid leakage. Lower back and leg pain visual analog scale (VAS) scores, Oswestry Disability Index (ODI) scores, modified by Fritz and Irrgang [10], and MacNab criteria (excellent, good, fair, poor) for evaluating disability and pain response were collected. Foraminal stenosis was classified with the Lee system (grade 0 to grade 3) [11] on the sagittal MRI as follows: Grade 0 refers to the absence of foraminal stenosis; grade 1 refers to mild foraminal stenosis showing perineural fat obliteration surrounding the nerve root in the two opposing directions (vertical or transverse). No evidence of morphologic change in the nerve root is shown. Grade 2 refers to moderate foraminal stenosis showing perineural fat obliteration surrounding the nerve root in the four directions without morphologic change in both vertical and transverse directions. Grade 3 refers to severe foraminal stenosis showing nerve root collapse or morphologic change. To define the presence of radiological instability, the criteria introduced by Ito were used [12]. In this context, a translation of 3 mm or more on a flexion–extension radiograph was considered to be indicative of instability. The Ethics Committee approved this study.

We documented POD [13,14] when the preoperative dysesthetic pain character changed or when the dysesthetic pain severity deteriorated and new dysesthetic symptoms presented. We used a POD grading system (grade 1 to grade 3) reported by Kim et al. [8] to assess the dorsal root ganglion (DRG) retraction injury during the foraminal and extraforaminal decompression. The POD was classified according to symptom severity, symptom continuity, and motor deficits, as follows: Grade 1 refers to dysesthesia due to compression before surgery. Minimal radiating pain similar to the preoperative pain in a compressed root-innervated region. Symptoms not concordant with but similar to the preoperative pain. Symptoms limited in the follow-up duration. Grade 2 refers to dysesthesia caused by DRG retraction. Moderate to severe dysesthetic pain or burning dysesthesia in a properly manipulated DRG-innervated region without definite motor deficits. Symptoms not concordant with the preoperative pain; other characteristics of dysesthesia present. Symptoms were usually limited in the follow-up duration. Grade 3 refers to dysesthesia due to DRG injury. Dysesthetic pain accompanied by motor deficits or atrophic change in a properly manipulated DRG-innervated region. Possibly permanent symptoms. Different from reflex sympathetic dystrophy. Patients with persistent preoperative motor deficits were not classified as having POD grade 3.

### 2.4. Statistical Analysis

Statistical analysis was performed by using PASW Statistics version 18.0 (SPSS Inc., Chicago, IL, USA). Continuous variables were expressed as means and standard deviations. Clinical VAS scores and modified ODI scores were measured preoperatively, 1 month postoperatively, 3 months postoperatively, and at the final follow-up, and MacNab criteria were assessed at the final follow-up; these were all reported by the patients and analyzed with paired *t*-tests. A *p*-value of < 0.05 was considered to indicate a significant difference.

## 3. Results

A total of 48 patients consisted of 21 males, and 27 females were included. The mean age was 67.6 ± 9.7 years (range: 41–87), the mean follow-up duration was 10.9 ± 4.9 months (range: 6–24), and the mean operation time was 73. 5 ± 6.4 min (range: 56–93) (Table 1). The foraminal stenosis grade of the treated spinal levels was grade 1 (*n* = 16, 33%), grade 2 (*n* = 20, 42%), grade 3 (*n* = 12, 25%), and grade 2 and above (*n* = 32, 67%) (Table 1).

Accompanying degenerative pathologies or previous operations at the operational level, which can deform intracanalicular structures, can negatively affect surgical outcomes, potentially leading to POD; adjacent segment disease (ASD) just below or above the interbody fusion level, degenerative spondylolisthesis, or isthmic spondylolisthesis (ISPL). There was one case of ASD (Figure 4), four cases of ISPL (Figure 5), and two cases of previous operations. Of the two patients who had previous operations, one underwent transforaminal endoscopic lumbar foraminotomy (TELF) at the same foramen to be decompressed, and the other underwent hemilaminectomy contralateral to the operating foramen. There was one case of postoperative segmental instability in the four patients with ISPL. The ASD patient had POD grade 2, and one of the previous operation patients (the TELF case) had POD grade 1 (Table 1).

Surgery-related complications, such as revision operations due to recurrent disc herniation or instability, incidental durotomy, segmental instability, hematoma, and POD did occur. Two cases of revision surgeries occurred due to recurrence, and the patients underwent transforaminal lumbar interbody fusion surgery. There were two cases of segmental instability, two cases of incidental durotomy, and one case of hematoma; however, we did not perform the revision surgery for the segmental instability and incidental durotomy (Table 2).

The total POD incidence was 12.5% (6 of 48); there were four patients with POD grade 1 (8.3%) and two with POD grade 2 (4.2%), but none with POD grade 3. The rate of the occurrence of POD grade 2 and above, which may be related to intraoperative DRG retraction injury, was revealed 4.2% (two with grade 2, none with grade 3) (Table 2).

There was a significant improvement in VAS scores. The mean scores and ranges preoperatively, 1 month postoperatively, 3 months postoperatively, and at the final follow-up were 7.2 ± 1.3 (5–9), 3.2 ± 0.7 (1–4), 2.4 ± 0.9 (1–3), and 2.3 ± 1.0 (1–3), respectively (*p* < 0.001 for all groups and follow-ups) (Table 3). There was a statistically significant improvement in ODI scores. The mean scores and ranges preoperatively, 1 month postoperatively, 3 months postoperatively, and at the final follow-up were 72.3 ± 9.5 (56–84), 32.6 ± 6.6 (24–62), 27.5 ± 5.2 (18–46), and 25.8 ± 5.5 (14–52), respectively (*p* < 0.001 for all groups and follow-ups) (Table 3). The MacNab criteria showed that there were 11 (23%), 35 (73%), and 2 (4%) excellent, good, and fair outcomes, respectively (Table 3).

## 4. Discussion

Currently, the targeted contralateral approach for foraminal and extraforaminal pathologies is not a usual technique in open and microscopic surgery, even in biportal endoscopic surgery. Moreover, there is a paucity of literature about this surgical approach.

Kim et al. published articles of the technical notes and case series describing the uniportal full endoscopic interlaminar contralateral approach for resolving contralateral lateral recess, foraminal, and extraforaminal lesions with technical developments [4,5,6]. Recently, Kim et al. reported a [8] retrospective cohort study comparing the clinical result and safety between the transforaminal endoscopic approach requiring DRG retraction and the interlaminar contralateral endoscopic approach without DRG retraction for foraminal and extraforaminal disease. The endoscopic interlaminar contralateral approach showed favorable clinical outcomes without significant difference from the transforaminal endoscopic approach. The results revealed that the endoscopic interlaminar contralateral approach could be a safe procedure with a lower postoperative dysesthesia rate and be an alternative in patients with an anatomically limited [9] L5-S1 level.

With these articles [4,5,6,8], the feasibility, safety, and good clinical outcomes of the ICELF procedures might be considered approved, but more substantial evidence and rationale are necessary for effective and safe application. For the rationale of the contralateral approach for the contralateral foraminal and extraforaminal stenosis over the unnecessary disruption of the anatomy of the ipsilateral unaffected side, we included the patients who have coexisting symptomatic lateral recess stenosis to be decompressed.

The DRG should be protected by working cannular retraction during transforaminal endoscopic lumbar foraminotomy. Compression of the DRG of the exiting nerve root may occur due to manipulation of the beveled endoscopic working cannula leading to temporary ischemia and POD [13,14].

The L5-S1 level has disadvantageous anatomical features during foraminal decompression with ICELF and TELF procedures, the inclination of disc space, and the long length of the facet joint overlapping the disc space [9]. Therefore, we included the patient with L5-S1 level pathologies to highlight the feasibility of ICELF procedures and investigated the POD to evaluate the approach-related safety concerned with DRG compression during foraminal and extraforaminal decompression.

Regardless of the lumbar level, additional risks of nerve root injury may arise due to increasing the operation time and more aggressive manipulation during endoscopic foraminal decompression in the severely stenotic neural foramen. A large proportion of the patients in this study have severe foraminal stenosis, 67% of grade 2 and grade 3 with Lee system [11], so more nerve root manipulation was expected during foraminal and extraforaminal decompression. Moreover, several patients showed increased operation time over 90 min.

Because of the large proportion of high-grade foraminal stenosis and L5-S1 level specific anatomical features, the incidence of POD of this study was expected to be higher than that in the case series by Lewandrowski et al. [15], which reported an incidence of 21.5% (11.6–33%) in multicenter frequency analysis. However, the total POD incidence was 12.5% (6 of 48) lower than the previously reported rate. Remarkably, the rate of the occurrence of POD grade 2 and above, which might be related to intraoperative DRG retraction injury, was revealed 4.2% (two with grade 2, none with grade 3), much lower than other published reports [15]. As a result, the POD occurrence and POD severity might be reduced by switching the surgical approach to ICELF while escaping the disadvantageous anatomical features of the L5-S1 level, even in the severe foraminal stenosis cases.

However, we could not prevent the POD occurrence fully; it may be because DRG manipulation was inevitable during dissection in the severely stenotic foramen even though direct DRG retraction was not needed with ICELF.

The extent and amount of facet joint removal have a significant effect on postoperative stability. Damage to the joint capsule and IAP has been shown to increase motion in both the sagittal and axial ranges [16]. On the other hand, superior facetectomy, even if bilateral, is well tolerated, with more than 80% of the spinal stiffness maintained with a bilateral partial superior facetectomy [17]. A preserving of the IAP and the joint capsule is critical to reducing postoperative instability.

Currently, microscopic, and biportal endoscopic techniques usually require two separate approaches to decompress the coexisting intra-and-extraspinal stenosis; the ipsilateral interlaminar approach and the paraspinal transforaminal approach. A significant amount of lateral facet joint, including IAP and isthmus removal, is usually needed to decompress the medial foraminal stenotic pathology with a paraspinal approach. At this point, adding an ipsilateral laminectomy may highly induce segmental instability or isthmic stress fracture.

Fortunately, we could minimize the IAP and joint capsule with the ICELF procedure. Medial foraminal enlargement with medial facet removal is inevitable to reach the extraforaminal region from the contralateral side, but only some part of medial IAP and less than half of medial SAP removal was performed (Figure 1b). Furthermore, the SAP was drilled in a tunnel shape with an endoscopic drill to preserve the joint capsule (Figure 2e). The results of the present study showed the facet joint preserving effect; only two patients (4.2%) showed instability after surgery, even in the patients, which included one case of ASD (Figure 4), four cases of ISPL (Figure 5), and two cases of revision surgeries. Two instability patients have not performed fusion operation during follow-up duration.

Although ICELF has several benefits, it should be performed in selective cases. With ICELF in the far-out region, pathologies at the inferior-ventral side of the ENR can be removed efficiently. However, enormous facet joint removal is necessary to reach the lesions on the superior-dorsal side of the ENR because the free use of optimized curved punches and endoscopic drills are limited to the uniportal small-diameter endoscope. The POD occurrence risk may also be increased due to the excessive manipulation of DRG. Therefore, far-out syndrome caused by the compression and entrapment of the L5 nerve root in the extraforaminal area between the hypertrophied L5 transverse process and sacral ala [18,19] is not an indication of ICELF.

This study has limitations. First, the number of ICELF cases was small, and the follow-up period was short. Second, the surgical outcomes could be different at other lumbar levels because only the L5-S1 level was included in the present study. A multicenter study with a larger sample size, longer follow-up period, and better design is required to confirm the findings of the current study.

## 5. Conclusions

ICELF is an effective procedure to resolve combined pathologies (lateral recess, foraminal, and extraforaminal degenerative pathologies) with one surgical approach at the L5-S1 level. ICELF may be a minimal DRG retracting and facet joint preserving procedure in foraminal and extraforaminal decompression. However, a certain level of endoscopic experience is required before attempting this technique, and this technique should be performed in selective patients.

## Figures and Tables

**Figure 1 jcm-10-01364-f001:**
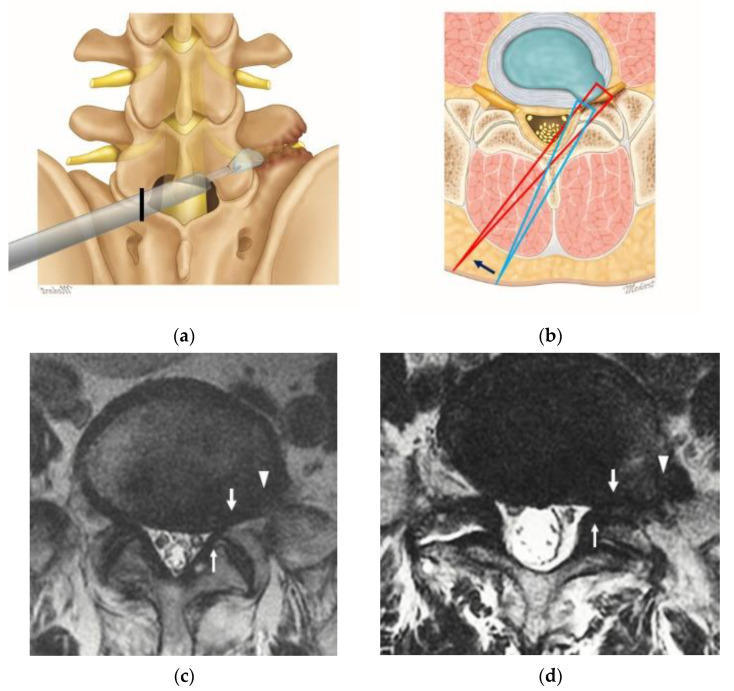
Illustration of the surgical approach for the coexisting intra-and-extraspinal root compression lesions at the L5-S1 level. (**a**) The surgical route of interlaminar contralateral endoscopic lumbar foraminotomy (ICELF) on a lumbar spinal model. The black line showed the skin incision point. (**b**) The trajectory of ICELF on the axial plane. Lateral recess decompression and medial foraminal opening were performed with an initial approaching trajectory (long blue triangle. The trajectory was then changed to a lower angle by levering the endoscope (long red triangle) to reach the foraminal and extraforaminal space. (**c**,**d**) Multifocal nerve root compression lesions at the intra-and-extra spinal canal are shown on the magnetic resonance imaging at the lateral recess (thin arrows), foraminal region (broad arrows), and extraforaminal region (arrowheads). All the indicated lesions could be resolved with one direction approach.

**Figure 2 jcm-10-01364-f002:**
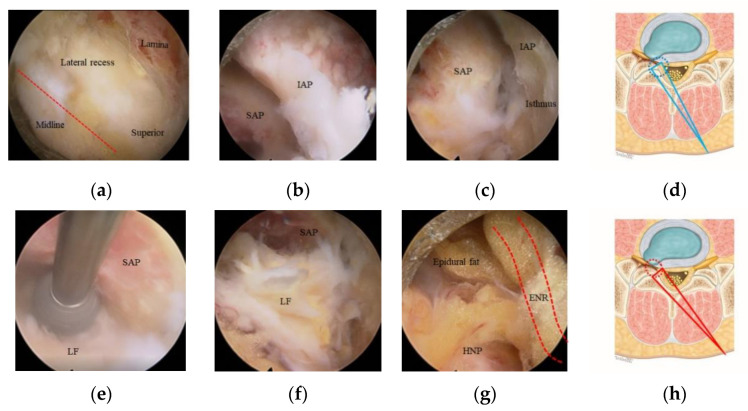
Intraoperative images of right-side uniportal endoscopic approach for left-side contralateral multifocal stenosis decompression at the L5-S1 level. (**a**) Contralateral sublaminar drilling was done to expose the lateral recess and foraminal entrance and then remove the hypertrophied ligamentum flavum (LF) using an endoscope with a 10-mm diameter. After lateral recess decompression, the endoscope was changed to one with a 7-mm diameter. (**b**) The medial part of the facet joint was exposed after LF removal. (**c**) After drilling the inferior articular process (IAP), the superior-medial part of the superior articular process (SAP) and attached foraminal LF were found. (**d**) The foramen exposure step (upper panel) was performed with an initially steeper trajectory (long blue triangle). The work area is outlined with a blue dotted circle. (**e**) The SAP was removed with an endoscopic drill in a tunnel shape to preserve the joint capsule, leaving the foraminal LF for nerve root protection. (**f**,**g**) The exposed foraminal LF was removed, the epidural fat tissue and foraminal herniated nucleus pulposus (HNP) were then found in the foraminal space. (**h**) The foramen opening step (lower panel) was done with a lower trajectory angle after endoscope levering (long red triangle). The work area is marked with a red dotted circle. ENR: exiting nerve root.

**Figure 3 jcm-10-01364-f003:**
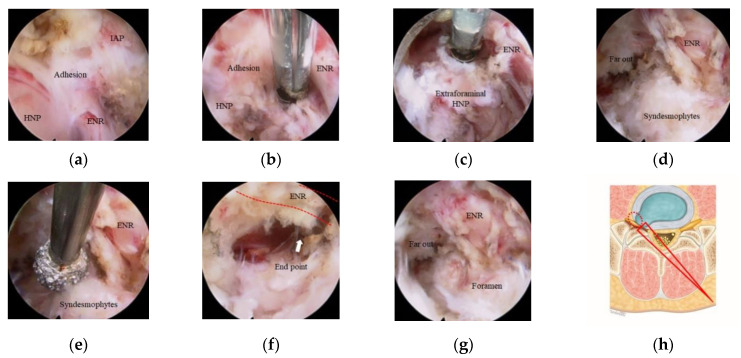
Intraoperative images during foraminal and extraforaminal decompression. (**a**) The compressed L5 exiting nerve root (ENR), herniated nucleus pulposus (HNP), and surrounding severe tissue adhesion were found. (**b**) Secure dissection along the ENR and foraminal discectomy were performed. (**c**) Extraforaminal discectomy was done without ENR retraction via an inferior-ventral foraminal tunnel formed after foraminal discectomy. (**d**) Decompression was done till the far-out region, but syndesmophytes were still compressing the ENR. (**e**) Syndesmophytes were removed by drilling along the ENR. (**f**) The endpoint (white arrow) is the far-out area where the exiting root began to curve downward with natural angulation. (**g**) Adequate decompression of the ENR in the foraminal and extraforaminal region was confirmed under endoscopic visualization. (**h**) The endoscopic trajectory (long red triangle) and work area (red dotted circle) of the were demonstrated on the axial illustration.

**Figure 4 jcm-10-01364-f004:**
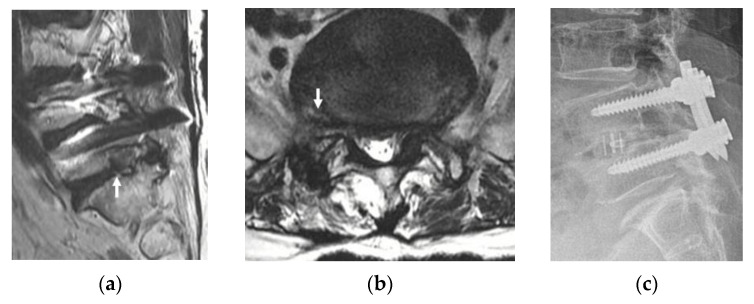
Case example. A 73-year-old female presented with radiating right leg pain and back pain. (**a**,**b**) Preoperative magnetic resonance imaging (MRI) showing lateral recess, foraminal, and extraforaminal stenosis at L5-S1 on the right (arrows). (**c**) Preoperative lateral radiograph showing previous interbody fusion at L4-L5 and a decreased L5-S1 foraminal window. (**d**,**e**) Adequate decompression of the lateral recess, foraminal, and extraforaminal lesions was confirmed on postoperative MRI (arrowheads). (**f**) Lateral radiograph at six months after surgery showing no definite spondylolisthesis or narrowing of the L5-S1 foramen. Transient grade 2 postoperative dysesthesia occurred.

**Figure 5 jcm-10-01364-f005:**
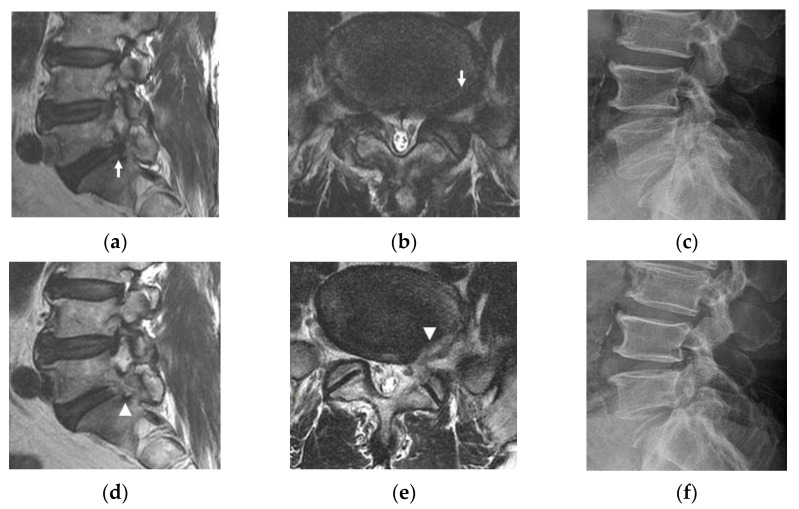
Case example. A 76-year-old female presented with left buttock pain and radiating left leg pain. (**a**,**b**) Preoperative magnetic resonance imaging (MRI) showing lateral recess stenosis with foraminal to extraforaminal HNP at L5-S1 on the left (arrows). (**c**) Spondylolysis without spondylolisthesis was found on preoperative lateral radiography. (**d**,**e**) Sufficient decompression of the lateral recess, foraminal to extraforaminal lesions was confirmed on postoperative MRI (arrowheads). (**f**) There was no spondylolisthesis progression on lateral radiography six months after the operation. Postoperative dysesthesia did not occur.

**Table 1 jcm-10-01364-t001:** Information of the patients.

Parameters	Patients (*n* = 48)
Age, years (range)	67.6 ± 9.7 (41–87)
Sex (male/female)	21/27
Follow-up period, months (range)	10.9 ± 4.9 (6–24)
Operation time, mins (range)	73. 5 ± 6.4 (56–97)
Foraminal stenosis (grade)	
Grade 0	0
Grade 1	16 (33%)
Grade 2	20 (42%)
Grade 3	12 (25%)
Grade 2 + 3	32 (67%)
Combined pathologies, *n* (complication)	
Isthmic spondylolisthesis	4 (*n* = 1, instability)
Adjacent segment disease	1 (*n* = 1, POD grade 2)
Previous operation	TELD (*n* = 1, POD grade 1), hemilaminectomy (*n* = 1, no POD)

Values are presented as means ± standard deviations or numbers (%). Foraminal stenosis was classified according to the Lee system [11]. POD: postoperative dysesthesia, TELD: transforaminal endoscopic lumbar discectomy.

**Table 2 jcm-10-01364-t002:** Postoperative complications.

Complication Type	Patient
Revision operation, *n*	2 (TLIF)
Segmental instability, *n*	2
Incidental durotomy, *n*	2
Hematoma, *n*	1
Postoperative dysesthesia, total, *n* (%)	6 (12.5%)
Grade 1	4 (8.3%)
Grade 2	2 (4.2%)
Grade 3	0

Postoperative dysesthesia was documented using the grading system. TLIF: transforaminal lumbar interbody fusion. Values are presented as numbers (%) for POD grading and numbers of involved patients.

**Table 3 jcm-10-01364-t003:** Clinical outcomes.

Parameter	Value	*p*-Value
Visual analog scale		
Preoperative	7.2 ± 1.3 (5–9)	-
Follow-up at 1 month	3.2 ± 0.7 (1–4)	<0.001 *
Follow-up at 3 months	2.4 ± 0.9 (1–3)	<0.001 *
Final follow-up	2.3 ± 1.0 (1–3)	<0.001 *
Oswestry Disability Index		
Preoperative	72.3 ± 9.5 (56–84)	-
Follow-up at 1 month	32.6 ± 6.6 (24–62)	<0.001 *
Follow-up at 3 months	27.5 ± 5.2 (18–46)	<0.001 *
Final follow-up	25.8 ± 5.5 (14–52)	<0.001 *
MacNab criteria at final follow-up, *n* (%)	Total, *n* = 48	
Excellent	11 (23%)	-
Good	35 (73%)	-
Fair	2 (4%)	-
Poor	0	-

Values are presented as means ± standard deviations. * A *p*-value of < 0.05 was considered statistically significant.

## Data Availability

All data generated or analyzed during this study are included in this published article.

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
