# Peer review of "The Novel Technique of Uniportal Endoscopic Interlaminar Contralateral Approach for Coexisting L5-S1 Lateral Recess, Foraminal, and Extraforaminal Stenosis and Its Clinical Outcomes"

_jcm, 2021, doi:10.3390/jcm10071364_

Round 1

Reviewer 1 Report

This manuscript is rated as a very good technical note, but not enough as a scientific paper.

Major issue

  1. Follow-up period

Follow-up period varies. 6 months look too short to evaluate surgical outcome.

Others

  1. Page 6, line 198; Please describe the full spelling of “POD”.

(Please remove “postoperative dysesthesia” in line 211.)

  1. Page 6, line 205; Please describe the detail of MacNab criteria.
  2. Page 6, line 207; Please describe or show Lee system for foraminal stenosis.
  3. Page 7, line 213; Please describe the detail of POD grading system.
  4. It is not necessary to have two decimal places of age, follow-up period, operation time, VAS and ODI
  5. What is the median and/or average of the final follow-up period?
  6. When did authors evaluate MacNab criteria?
  7. How did authors manage incidental durotomy?
  8. For outcome measurement, revision operation is judged as poor or should be excluded. What do authors think?
  9. What is the prognosis of POD in this series?

Author Response

Response to reviewer

Reviewer 1

Comments and Suggestions for Authors

This manuscript is rated as a very good technical note, but not enough as a scientific paper.

: We appreciate to reviewer's detailed comments.

Major issue

  1. Follow-up period

Follow-up period varies. 6 months look too short to evaluate surgical outcome.

 : Yes, six months follow-up duration after surgery is short to evaluate the sufficient clinical with radiological outcomes, especially in operations with extensive bone removal and fusion. However, endoscopic neural decompression surgery, preserving the normal functional structures, may be investigated with surgical outcomes after six months. We planned to include the patients who had at least six months follow-up duration; however, few patients had follow-duration around six months, and the mean follow-up duration was approximately 11 months. A longer follow-up study is required to confirm the findings of the current study; we revised the manuscript with these responses in the discussion, limitation part (Page 13, Line 391, 394). Our team has been planning the longer follow-up and larger sample size study to overcome the limitations mentioned in the limitation.

Others

  1. Page 6, line 198; Please describe the full spelling of “POD”. (Please remove “postoperative dysesthesia” in line 211.): We corrected them
  1. Page 6, line 205; Please describe the detail of MacNab criteria.: We corrected it (excellent, good, fair, poor)
  2. Page 6, line 207; Please describe or show Lee system for foraminal stenosis.

We add the Lee system as follows:

Grade 0 refers to the absence of foraminal stenosis

Grade 1 refers to mild foraminal stenosis showing perineural fat obliteration surrounding the nerve root in the two opposing directions (vertical or transverse). It in-volves contact with the superior and inferior portions of the nerve root or anterior and posterior portions of the nerve root. No evidence of morphologic change in the nerve root is shown.

Grade 2 refers to moderate foraminal stenosis showing perineural fat obliteration surrounding the nerve root in the four directions without morphologic change in both vertical and transverse directions.

Grade 3 refers to severe foraminal stenosis showing nerve root col-lapse or morphologic change.

  1. Page 7, line 213; Please describe the detail of POD grading system.

We add the POD grading system as follows:

Grade 1: Dysesthesia due to compression before surgery. Minimal radiating pain similar to the preoperative pain in a compressed root-innervated region. Symptoms not concordant with but similar to the preoperative pain. Symptoms limited in the follow-up duration.

Grade 2: Dysesthesia caused by DRG retraction. Moderate to severe dysesthetic pain or burning dysesthesia in a properly manipulated DRG-innervated region without definite motor deficits. Symptoms not concordant with the preoperative pain; other characteristics of dysesthesia present. Symptoms usually limited in the follow-up duration.

Grade 3: Dysesthesia due to DRG injury. Dysesthetic pain accompanied by motor deficits or atrophic change in a properly manipulated DRG-innervated region. Possibly permanent symptoms. Different from reflex sympathetic dystrophy. Patients with persistent preoperative motor deficits were not classified as having POD grade 3.

  1. It is not necessary to have two decimal places of age, follow-up period, operation time, VAS and ODI

: Right, one decimal is enough. We corrected them.

  1. What is the median and/or average of the final follow-up period?

We included the patients who had a minimum of six months of the follow-up period, and the final follow-up duration was used to analyze the average value. Therefore, described mean follow-up duration refers to the final follow-up period.

Follow-up period, months (range)

10.9 ± 4.9 (6-24)

  1. When did authors evaluate MacNab criteria?

: MacNab criteria were measured at each follow-period, and we used the final follow-up value.

: We described this point in Table 3.

  1. How did authors manage incidental durotomy?

: Occurred durotomy size was small, so the durotomy hole was sealed with a TACHOSIL patch under endoscopic view without conversion to open surgery. Then, patients were managed with 2 or 3 days of bed rest according to the symptoms and durotomy size.

  1. For outcome measurement, revision operation is judged as poor or should be excluded. What do authors think?

: The ICELF technique has benefits in the revision of foraminal surgery because the endoscope could advance to the foraminal and extraforaminal space through the virgin tissues at the medial foraminal entrance. To highlight these approach-related benefits, we included the patients who had transforaminal surgery previously.

  1. What is the prognosis of POD in this series?

: The patients who had POD were treated with conservative management, some pregabalin, and nerve root block. The dysesthesia symptoms were resolved within six weeks in all POD occurred patients.

Reviewer 2 Report

I think the manuscript is well written and very interesting.
However, I have a few concerns.
The first is the inclusion of cases with a short follow-up period. I also think that the small number of cases, as mentioned in the limitations, is a major problem.

This article describes a new technique for Uniportal Endoscopic Interlaminar Contralateral Approach for Coexisting L5-S1 Lateral Recess, Foraminal, and Extraforaminal Stenosis. 
The inclusion and exclusion criteria are also carefully written and useful.
The results are good and I think it will be useful information for many spine surgeons.

Author Response

Response to reviewer

Reviewer 2

Comments and Suggestions for Authors

I think the manuscript is well written and very interesting.
However, I have a few concerns.

The first is the inclusion of cases with a short follow-up period. I also think that the small number of cases, as mentioned in the limitations, is a major problem.

: Yes, six months follow-up duration after surgery is short to evaluate the sufficient clinical with radiological outcomes, especially in operations with extensive bone removal and fusion. However, endoscopic neural decompression surgery, preserving the normal functional structures, may be investigated with surgical outcomes after six months. We planned to include the patients who had at least six months follow-up duration; however, few patients had follow-duration around six months, and the mean follow-up duration was approximately 11 months. A longer follow-up study is required to confirm the findings of the current study; we revised the manuscript with these responses in the discussion, limitation part (Page 13, Line 391, 394). Our team has been planning the longer follow-up and larger sample size study to overcome the limitations mentioned in the limitation.

This article describes a new technique for Uniportal Endoscopic Interlaminar Contralateral Approach for Coexisting L5-S1 Lateral Recess, Foraminal, and Extraforaminal Stenosis. 
The inclusion and exclusion criteria are also carefully written and useful.
The results are good and I think it will be useful information for many spine surgeons.

: We appreciate to reviewer's positive comments.
